# Polarisation of decayless kink oscillations of solar coronal loops

Sihui Zhong [1], Valery M. Nakariakov [1,2] ✉, Dmitrii Y. Kolotkov [1,3], Lakshmi Pradeep Chitta [4], Patrick Antolin [5], Cis Verbeeck[6] & David Berghmans [6]

Decayless kink oscillations of plasma loops in the solar corona may contain an answer to the enigmatic problem of solar and stellar coronal heating. The polarisation of the oscillations gives us a unique information about their excitation mechanisms and energy supply. However, unambiguous determination of the polarisation has remained elusive. Here, we show simultaneous detection of a 4-min decayless kink oscillation from two non-parallel lines-of-sights, separated by about 104°, provided by unique combination of the High Resolution Imager on Solar Orbiter and the Atmospheric Imaging Assembly on Solar Dynamics Observatory. The observations reveal a horizontal or weakly oblique linear polarisation of the oscillation. This conclusion is based on the comparison of observational results with forward modelling of the observational manifestation of various kinds of polarisation of kink oscillations. The revealed polarisation favours the sustainability of these oscillations by quasi-steady flows which may hence supply the energy for coronal heating.

The outermost, fully-ionised and magnetically-dominated part of the atmosphere of the Sun, the corona, attracts our attention as the birthplace of extreme events of space weather, such as flares and coronal mass ejections[1]. The corona is a natural plasma physics laboratory allowing for high-resolution study of basic phenomena which are of interest for various astrophysical, geophysical and laboratory plasma physics applications. The corona offers us several enigmatic puzzles, such as the rapid release of the magnetic energy, and heating of the plasma to temperatures about three orders of magnitude higher than the surface temperature. One of the most rapidly developing avenues of modern coronal physics is the study of magnetohydrodynamic (MHD) wave phenomena, see, e.g., ref. 2. Coronal waves are of specific interest as possible agents of coronal heating, see, e.g., ref. 3, seismological probes of the coronal plasma structures e.g., refs. 4–7, and an important element of flaring energy releases, e.g., ref. 8. A ubiquitous phenomenon are transverse, or kink,

oscillations of coronal plasma loops[9], which appear in two different regimes. Large-amplitude rapidly-decaying kink oscillations are normally excited by displacements of the loops from the equilibrium, caused by low-coronal eruptions[10], and decay by resonant absorption (e.g., ref. 11) and/or by Kelvin–Helmholtz instability (KHI, e.g., refs. 12,13). The nature of another type, low-amplitude decayless kink oscillations[14–17] is subject to intensive ongoing studies. The mechanisms which could counteract the oscillation damping are random footpoint movements, e.g., refs. 18–20 and quasi-stationary coronal, chromospheric or photospheric flows, e.g., refs. 21–23, or their combination[24]. Alternative mechanisms for the apparent decayless behaviour are the Kelvin-Helmholtz instability[25] and a pattern of interference fringes[26]. In addition, field-aligned flows such as coronal rain could be accompanied by decayless kink oscillations[27,28].

In contrast with large-amplitude decaying kink oscillations which are relatively rare and therefore cannot contribute to coronal heating,

[1]Centre for Fusion, Space and Astrophysics, Physics Department, University of Warwick, Coventry CV4 7AL, UK. [2]Centro de Investigacion en Astronomía, Universidad Bernardo O'Higgins, Avenida Viel 1497, Santiago, Chile. [3]Engineering Research Institute 'Ventspils International Radio Astronomy Centre (VIRAC)' of Ventspils University of Applied Sciences, Inzenieru iela 101, Ventspils LV-3601, Latvia. [4]Max Planck Institute for Solar System Research, D-37077 Göttingen, Germany. [5]Department of Mathematics, Physics and Electrical Engineering, Northumbria University, Newcastle Upon Tyne NE1 8ST, UK. [6]Solar-Terrestrial Centre of Excellence - SIDC, Royal Observatory of Belgium, Ringlaan -3- Av. Circulaire, Brussels 1180, Belgium. ✉e-mail: V.Nakariakov@warwick.ac.uk

the ubiquity of decayless kink oscillations makes them an important process for the coronal energy balance[29]. On one hand, the nature and importance of damping mechanisms such as resonant absorption and KHI, are still under debate. In particular, the concept of coronal heating by steady state KHI turbulence requires higher-than the observed amplitudes of decayless oscillations to sustain the quiet solar corona[30]. On the other hand, as it has been shown in[29], decayless kink oscillations can represent a significant energy input in the loop. In other words, their relatively low amplitudes are not representative of the total amount of energy converted by this process. In this context, a crucial question is the mechanism which counteracts the oscillation damping and hence is responsible for the energy supply to the corona. One of the key indicators of the oscillation excitation mechanism and hence of such an energy supply mechanism is the oscillation polarisation. For example, decaying kink oscillations polarised in the horizontal and vertical directions have been observed to be excited by a non-radial eruption of a magnetic flux rope[31]. Likewise, the coronal rain has been shown to lead to vertically polarised transverse oscillations of a loop in 2.5D MHD simulations[28]. In the decayless regime, the self-oscillatory model pictures a linear polarisation[22,23], while the randomly-driven model depicts a random polarisation[19,32].

In a few very rare cases when the top segments of the oscillating loops were parallel to the line-of-sight (LoS), the polarisation of large-amplitude decaying kink oscillations could be determined[33,34]. Another potential opportunity appears if imaging observations are supplemented with spectroscopic observations[35]. However, only stereoscopy, i.e., observations from different vantage points would allow one to study 3D geometry of kink oscillations properly. So far, there has been only one attempt to estimate the polarisation plane stereoscopically, for a large-amplitude decaying regime[36]. The oscillation with a period about 10.5 min, damping time about 17 min, and initial amplitude of about 4 Mm was detected with the Extreme Ultraviolet Imager (EUVI) instruments on board the two Solar TErrestrial RElations Observatories (STEREO). The instruments had a temporal resolution of 2.5 min, and the pixel sizes 1100 km and 1240 km, respectively. The two STEREO spacecraft were separated by 15.4 degrees. Unfortunately, in that observation the footpoints of the oscillating loop were behind the limb for both vantage points. Thus, the stereoscopic reconstruction was based on the observations of footpoints three days later than the oscillation event, when the loop was not visible. An additional information used in this study was the perturbation of the emission intensity in the loop, caused by the variation of the column depth in the optically thin regime. Assuming that the oscillation was linearly polarised, it was deduced that the oscillation was polarised rather

horizontally. This result is consistent with the excitation of a large-amplitude decaying kink oscillation by an impulsive displacement[10].

For decayless oscillations, until the commissioning of high-resolution EUV imagers of the new generation, it has not been possible to determine the polarisation sense and constrain the mechanism by which the energy is transferred to the corona to sustain the oscillation. In particular, one would expect that random footpoint motions result in a randomly varying polarisation.

Here, we report simultaneous high-resolution observations of a decayless kink oscillation of a coronal loop with two non-parallel LoS with the Atmospheric Imaging Assembly (AIA[37]) on board the Solar Dynamics Observatory (SDO[38]), and the Extreme Ultraviolet Imager (EUI[39]) on board the recently launched Solar Orbiter (SolO[40]). It allows us to identify the type of polarisation of the observed oscillation, and discuss its implications for constraining the energy supply mechanism.

## Results

### Spacecraft constellation

During 09:19 to 10:34 UT on April 1st 2022, when SolO was near the perihelion of its orbit, HRIEUV (EUI's High Resolution Imager which observes the Sun in the 174 Å passband[39]), detected decayless kink oscillations in a bundle of loops on the solar disk near the limb. The oscillating loops were situated at the outskirt of NOAA active region (AR) 12976 (Fig. 1) of the $\beta$ Hale class. Thanks to high resolutions down to 123 km per pixel, the oscillations can be well seen by eye in the movie constructed by a sequence of images (see Supplementary Movie 1a). Meanwhile, AIA took images of the same bundle of loops from a different LoS, see Fig. 1b. As the spatial resolution of AIA images (435 km/pixel) is about 3.5 times HRIEUV's, the low-amplitude oscillations recorded by AIA are not resolvable by naked eye in the movie. However, the oscillations were clearly detected in a movie processed with the motion magnification technique[41,42] (Supplementary Movie 1b). No flares or CMEs are reported in this AR during the observations. During the observation, the distance from the Sun for SolO is 0.350 AU, and that for SDO is 0.999 AU Therefore, the light travel time difference between two spacecrafts was 325.5 s (5.4 min). The Carrington longitude and latitude of centres of the field-of-view (FoV) for HRIEUV and full-disk AIA/SDO data roughly [135.5 degrees, 2.3 degrees] and [31.8 degrees, −6.5 degrees] respectively, see their location in Fig. 1a. The angle separation between the two LoS was 103.95 degrees.

### Oscillation properties

As shown in Supplementary Movie 1, the oscillating loop bundle consists of multiple strands which perform a collective transverse

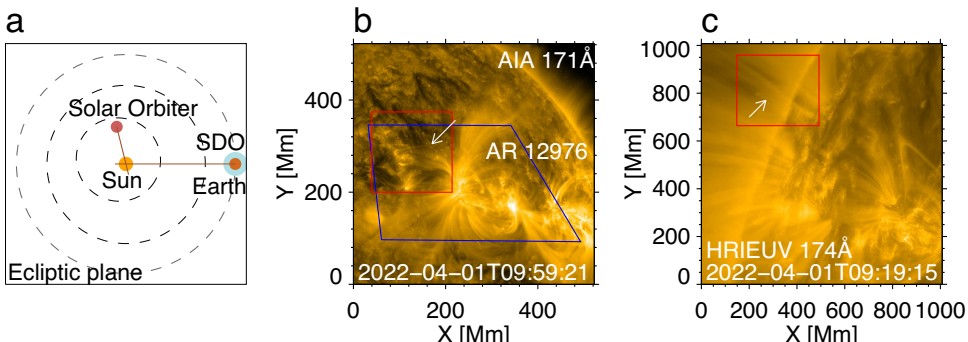

**Fig. 1 | The location of two spacecrafts during the observations and images of the regions of interest. a** The position of Solar Dynamics Observatory (SDO) and the Solar Orbiter (SoLO) in Heliocentric Earth Ecliptic (HEE) coordinate system. The space missions are indicated by smaller circles. The grey dash path are the orbits of Mercury, Venus, and Earth (marked by the blue circle) from inner to outward direction. The brown straight lines indicate the Line-of-Sight (LoS) of HRIEUV/SoLO and AIA/SDO. **b** An AIA 171 Å image indicating the region of interest (the red box). Oscillating loops are denoted by the white arrow. The blue quadrilateral indicates the FoV of HRIEUV, projected onto the AIA image plane. **c** An HRIEUV 174 Å image indicating the region of interest by the red box. Source data are provided as a Source Data file.

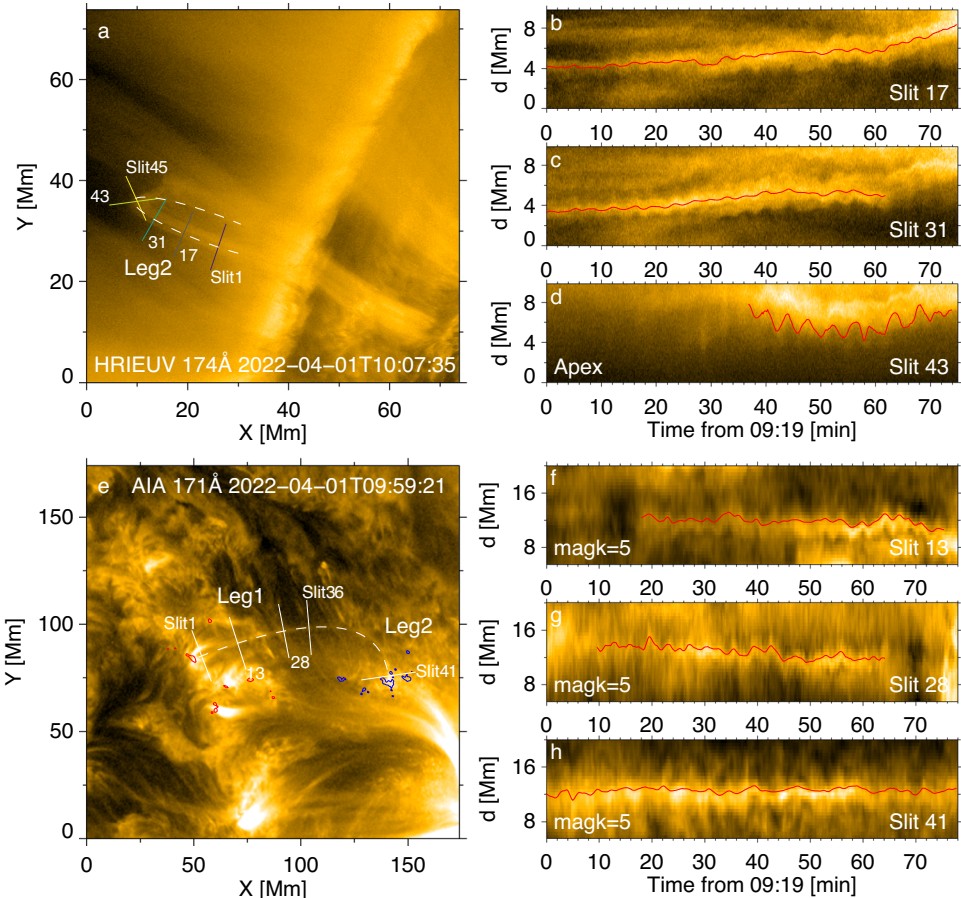

**Fig. 2 | Decayless kink oscillations of the analysed loop bundle.** The oscillating loop shown in HRIEUV (**a**) and AIA (**e**) images is best-fitted with truncated ellipses (white dashed curves). The loop legs are indicated by labels, Leg1 and Leg2. In panel **e**, the LoS magnetogram is over-plotted to show the magnetic connectivity (positive in red and negative in blue). The slits across the loop are used to make time–distance plots shown on the right (panels **b**–**d**, **f**–**h** for HRIEUV and AIA,

respectively). In the time--distance plots, the corresponding slit index numbers are denoted, and the displacements of loop centre/boundary are marked by the red curves. For the AIA data set, the time--distance plots were made with the data processed with the motion magnification coefficient *magk* = 5. Source data are provided as a Source Data file.

oscillation. Therefore, rather than analysing a specific strand, we treat the bundle as a single entity. From HRIEUV's view (Fig. 2a and Supplementary Movie 1a), only one leg and the apparent apex of the oscillating loop bundle can be clearly identified. The footpoints are not clearly seen, and the other leg is overlaid by another loop coming into LoS in the second half of the analysed time interval. In AIA images (Fig. 2e and Supplementary Movie 1b, c), the footpoints and one leg are well visible, but some segments of the oscillating bundle are overlapped with bright plasma flows in the background most of the time. The HMI LoS magnetogram shows that the loops forming the bundle connect a pair of photospheric magnetic patches of opposite polarities, see the red and blue contours overlaid on a coronal AIA EUV image in Fig. 2c. Time–distance maps were constructed for some particular loop segments whose transverse oscillations were well resolvable, with slits put across the loop (Fig. 2b–d, f–h). Both HRIEUV and AIA time–distance maps demonstrate transverse oscillatory patterns. The oscillations appear to be almost monochromatic, and last for tens of cycles. The oscillation amplitudes do not show systematic decay or growth, which is typical for the decayless regime (e.g., ref. 43).

The oscillatory patterns are clearly evidenced in time–distance maps constructed for various perpendicular slits. To make the narrative easier, the loop bundle is divided into three segments – Leg 1, (apparent) Apex, and Leg 2. For oscillations detected by HRIEUV (Fig. 2b–d), the apparent displacement amplitude is seen to increase towards the apex. Moreover, detrended oscillations of different

segments of the loop appear to be in phase with each other (Fig. 3a). This behaviour is consistent with the fundamental kink mode (see ref. 44). This conclusion is also confirmed by the in-phase oscillatory patterns detected with AIA in Leg 1, Apex, and Leg 2 (Fig. 3d). Note that oscillations in AIA data set are less clear due to the lower resolution and the abrupt interruption of the background evolution during the observation. In particular, slits 13, 28, and 41 show the most clear oscillation examples in the corresponding loop segments. Only those slits which do not show significant deviations from a quasi-oscillatory pattern are accepted for further analysis. Figure 3c displays Fourier power spectra of the oscillatory patterns made for different slits with the use of HRIEUV. As the analysed oscillation is the fundamental mode, i.e., all segments of the loop oscillate in phase, we may improve the signal-to-noise ratio by averaging the spectra over a particular segment, see Fig. 3b. The oscillations detected in legs and at the apex have the same power peak at about 4 min, which further corroborates the detection of the fundamental mode. Likewise, the AIA data demonstrate the domination of 4-min oscillations in all segments of the loop (Fig. 3e, f). The nature of the minor peaks at about 8–9 min (HRIEUV) or 6–12 min (AIA) is not clear, but it is not relevant to the analysis of 4-min oscillations.

The determination of the oscillation polarisation is based on the estimation of the phase difference between oscillations detected with HRIEUV and AIA, i.e., from two different vantage points (see methods subsection data processing and analysis). As the phase comparison

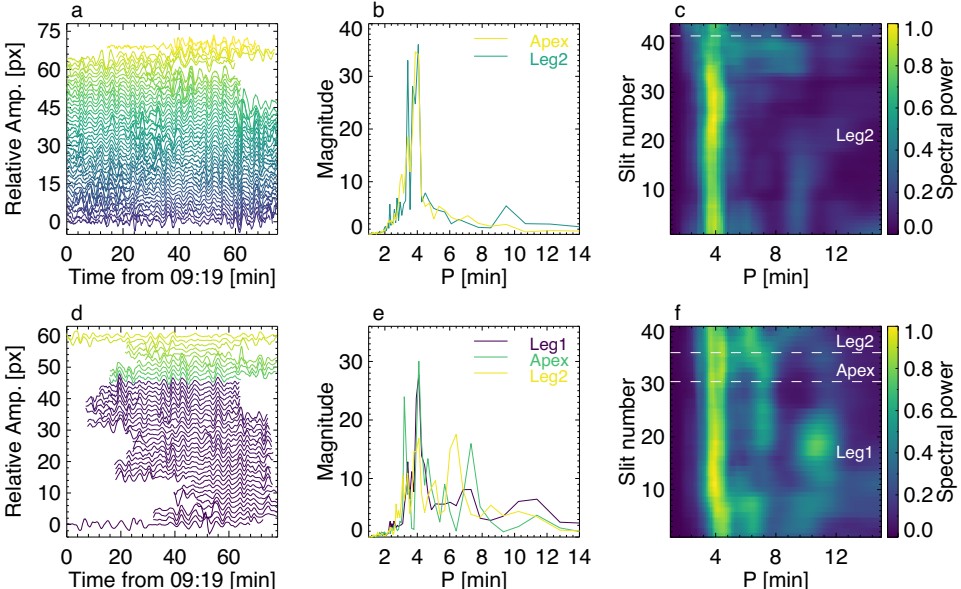

**Fig. 3 | Analysis of observed kink oscillations.** Oscillatory transverse displacements of different segments along the analysed loop (panels **a** and **d**, for HRIEUV and AIA, respectively), and their Fourier power spectra (panels **b**, **c**, **e** and **f**). Panels **a** and **d** show the detrended oscillations signals, obtained by subtracting the trend from original displacements determined by the time--distance maps. The trend is obtained by smoothing the original signals with a window longer than the oscillation period. The colour scheme indicates the slit index numbers, with the dark blue corresponding to smaller numbers. Panels **b** and **e**: Spectral power averaged over a particular segment of the oscillating loop. Panels **c** and **f**: Fourier power spectra of the transverse displacements of different slits, with the colour scheme showing the spectral power normalised to the maximum. Source data are provided as a Source Data file.

requires the adjustment of the time scales, the light travel time correction is applied to the AIA signal, allowing us to account for the difference in the distances from the Sun to the spacecraft. In addition, the instantaneous amplitudes are normalised by their envelope, to remove the amplitude modulation. In order to minimise the effect of background noise, the transverse displacements are averaged over five neighbouring slits (around 10 Mm wide segment) near the apparent apex in both data sets. Eleven oscillation cycles were detected in the common time interval. As shown in Fig. 4a, original signal-pairs seem to be approximately in anti-phase. The cross-correlation coefficient calculated with different time lags peaks at a time lag of 120 s, almost exactly at half of the oscillation period (4 min). Thus, the phase shift is about 180 degrees. Also, we constructed a phase portrait (also called a hodogram, Fig. 4b) of the normalised oscillatory patterns detected with AIA and HRIEUV (see methods subsection data processing and analysis). The hodogram demonstrates that, in general, the shape of most of the mutual oscillation cycles is approximately elliptical with a high eccentricity and negative inclination. Given that the amplitudes are normalised, the eccentricity of the ellipse coincides with the length of its minor axis. To simplify the measurement, the pattern is rotated clockwise with 45 degrees so that the minor axes of the ellipses related to various oscillation cycles appear in the horizontal direction, as displayed in panel c. The histogram of the distribution of the minor axis lengths approaches a Gaussian. The average length of the minor axes, $\sigma$, is estimated by the width (standard deviation) of the best fitted Gaussian, resulting in $\sigma = 0.19$. Thus the ratio of the major and minor axes of the hodogram ellipses is $0.19/0.7 = 0.27$, where 0.7 is the half width of the major axis.

To get rid of the undesired noisy fluctuations beside the 4-min oscillation, we filter out the time-series data with peak frequency with a 30-s (0.000506 Hz) window, resulting in cleaner oscillatory signals, as displayed in Fig. 4d–f. Similar to the original signals, the filtered oscillatory signals detected with AIA and HRIEUV appear to be out of phase with each other. The cross-correlation coefficient peaks at a time lag of −0.48 periods, indicating a phase shift of 174 degrees. As shown in Fig. 4e, f, the hodogram trajectory of each cycle is a narrow ellipse

with $\sigma = 0.08$ and an elongation ratio of 0.11. Repeating the above analysis with the signals averaged over 7 slits near the apex, similar results are obtained: the oscillatory signal detected with AIA and HRIEUV are in anti-phase, and the hodogram trajectories are highly elliptical, with $\sigma = 0.05$ in the minor axis.

## Polarisation signatures in forward modelling

With the analytical model detailed in methods subsection the geometrical model, we reproduce kink oscillations of a 3D loop with different types of polarisation, as seen in the LoS of HRIEUV and AIA (see Figs. 5 and 6). Projecting the modelled loop onto the planes-of-sky (PoS) of HRIEUV and AIA (Fig. 5), the simulated 2D images are seen to reproduce well the observed geometry. The dynamics of the modelled loop, manifesting harmonic kink oscillations of different polarisation (i.e., horizontal, vertical and oblique linear with respect to the loop's plane, and also circular, and elliptical) in the HRIEUV (left) and AIA (right) LoS, is shown by Supplementary Movie 2. For all types of polarisation seen in the simulated AIA LoS, oscillations of all loop segments are in phase, as it is expected for the fundamental kink mode and is consistent with observations (see Fig. 3). Likewise, kink oscillations seen in the modelled HRIEUV LoS in Leg 2 and near the loop apex are also in phase for all types of the polarisation (Leg 1 is not well seen in HRIEUV observations, hence cannot be used for comparison).

On the other hand, each type of polarisation is expected to have its own characteristic signatures, specific for the considered combination of AIA and HRIEUV observations. To reveal these distinct signatures, we apply the same time−distance analysis to the synthetic data sets as it has been done for revealing the observed oscillation properties in results subsection oscillation properties. Namely, we select segments of the modelled loop projected on the planes-of-sky (PoS), similar to those used in the observations (see Fig. 2 and the green and red slits in Fig. 5). In particular, Fig. 6 shows synthetic time−distance maps with the modelled kink oscillatory patterns of different polarisation seen with two LoS, taken at the slit situated at Leg 2 in the synthetic HRIEUV image and at Leg 1 in the synthetic AIA image respectively. Extracting the loop boundary displacements

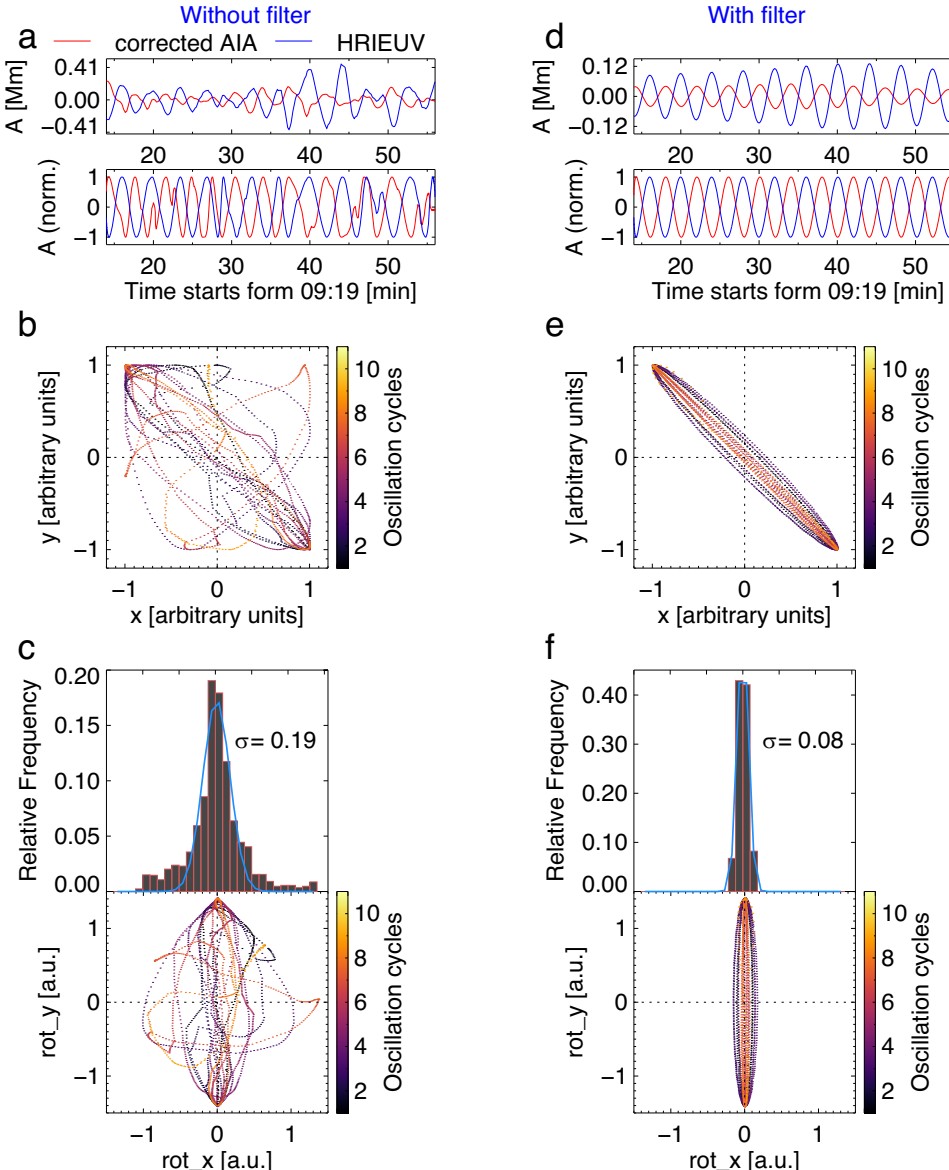

**Fig. 4 | Phase shifts between the kink oscillations detected with HRIEUV and AIA.** The signals are averaged over 5 slits near the apex. Panels **a**–**c** show original signals. Panels **d**–**f** show the signals filtered out in the vicinity of the 4-min period with a spectral window of 30 s. The AIA signals are shifted forwards in time by 325.5 s to compensate the light travel time difference. In panels **a** and **d**, the signals are shown as a function of time. In the subpanels, both original and normalised signals, which get rid of the amplitude modulation, are demonstrated. Panels **b** and **d**, display hodograms of the oscillatory signals. The colour code indicates oscillation cycles from 1 to 11. Panels **c** and **f** demonstrate the estimation of the elongation ratio with the Gaussian width $\sigma$. Here, a.u. stands for arbitrary units. Source data are provided as a Source Data file.

obtained in two LoS from the modelled time–distance maps (see Fig. 6f–j), we calculate their phase difference and simulate hodograms (see Fig. 6k–o, p–t, respectively). In the case of horizontal and oblique polarisations, the oscillatory AIA and HRIEUV signals appear to be in anti-phase, and the resulting phase portrait shows a narrow structure with a negative gradient (panels p and r). For vertical polarisation, the AIA and HRIEUV signals are in phase, also resulting in a narrow phase trajectory but with a positive gradient (panel q). For the circular polarisation, oscillations taken in two LoS are about −120 degrees out of phase with each other, leading to an elliptical trajectory in the hodogram, with the ratio of the major to minor axes about 0.6 (panel s). Similarly, in the case of the elliptical polarisation, oscillations are about −144 degrees out of phase, making a narrower ellipse with the axes ratio of 0.45 in the phase portrait. Moreover, the distribution of phase trajectories along the minor axis is different for

different polarisations. As shown in Fig. 7, in the linear polarisation (panels a–c) including the horizontal, vertical, and oblique ones, the histograms of the phase trajectories along the minor axes are single-peaked with very narrow widths. It means most of the trajectories are localised near the axis origin. Circular and elliptical polarisations are manifested by bimodal distributions with double peaks at the sides of the minor axes (see panels d, e). Thus, we consider the tim–phase difference between AIA and HRIEUV oscillatory signals and the corresponding hodogram shapes together with the localisation of phase trajectories along the minor axes as distinct characteristic signatures of different kink polarisation types, to be compared with observations.

In addition, for the oblique and elliptical polarisations in the AIA LoS, oscillation profiles manifest distortions from a harmonic shape, so that the waveforms of the AIA signals in Fig. 6m and o are asymmetric

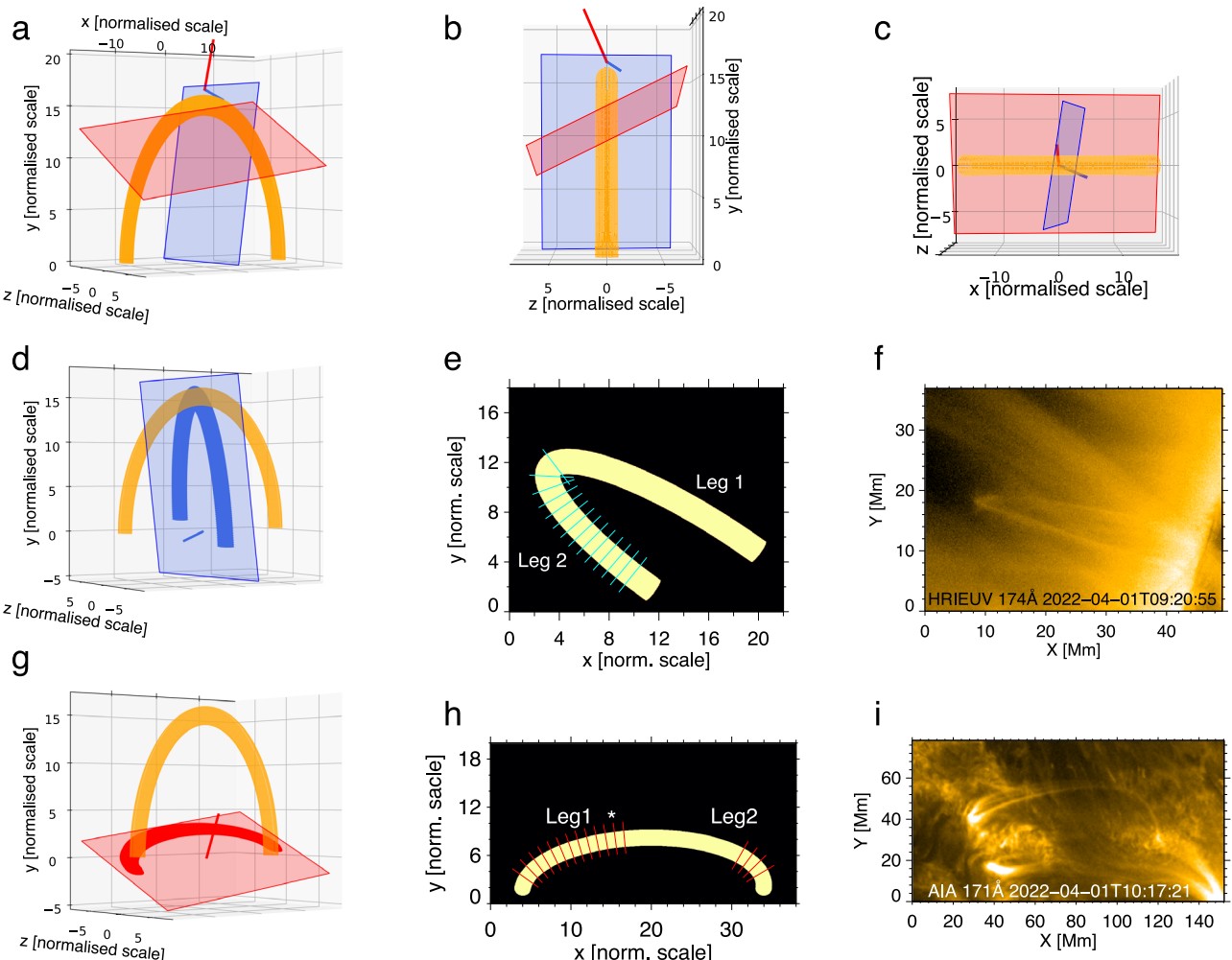

**Fig. 5 | Modelling an oscillating coronal loop observed in two non-parallel LoS mimicking that of the HRIEUV and AIA observations. a–c** The modelled loop, LoS, and plane-of-sky (PoS), seen in different views. The blue and red normal vector and the plane indicate the LoS and corresponding PoS of HRIEUV and AIA images, respectively. **d–i** Projection of the 3D loop onto a particular PoS (**d**, **g**) and the resulting simulated image (**e**, **h**), with the observational image (**f**, **i**) for comparison. Panels **d–f** are for the modelled HRIEUV LoS, and Panels **g–i** for AIA. The slits cover the segments similar to those analysed in the observations. The stars denoted the slits which are used to make time-distance maps in Fig. 6. The scales in the simulated 3D data and 2D images are normalised to the minor radius of loop (unit length).

with respect to the horizontal and vertical axes, respectively. This, in turn, leads to the skewness of the corresponding phase trajectories in hodograms (Figs. 6r, t and 7c, e), which however remains small. As the kink oscillations are assumed to be linear in the model (see Eqs. (4) and (5)), these waveform distortions are attributed to projection effects only. Similar skewed waveforms have been observed in, e.g., decaying kink oscillations of a flux rope[45].

### Determination of polarisation
Comparison of Figs. 4 and 6 allows us to exclude the possibility of the vertical polarisation in the observed decayless event, as the in-phase behaviour of the simulated AIA and HRIEUV signals and the phase portrait with a positive gradient are not consistent with observations. The other four types of polarisation result in the phase portraits of the AIA and HRIEUV signals with a negative gradient, and thus are in general agreement with the observed oscillation properties. Moreover, the observed phase delay of 174 degrees between oscillations in two LoS is broadly consistent with the horizontally, obliquely, and elliptically polarised oscillation models, taking the observational and data analysis uncertainties into account. However, only models with horizontal and oblique linear polarisations (see Fig. 7a, c) successfully reproduce the localisation of the majority of phase trajectories near

the origin in the hodogram in Fig. 4. In other words, the observed Gaussian distribution of the minor axes in Fig. 4c can be caused by the horizontal or oblique linear polarisation of oscillations only, whereas the finite width of the obtained Gaussian peak can be attributed to noise of the instrumental or another origin.

The further discrimination between horizontal and oblique linear polarisations can be based on a skewed hodogram shape of the latter, caused by the distortion of the oscillation profiles from a harmonic shape due to projection effects as discussed in methods subsection the geometrical model. However, as this effect is found to be rather minor in modelling, it is most likely to be obscured by noise in Fig. 4a–c. Suppression of noise by narrow-band Fourier filtering (in Fig. 4d–f), described in results subsection oscillation properties, cannot account for this skewness due to the essentially harmonic nature of the Fourier transform. Hence, we conclude that the horizontal and oblique linear polarisations of decayless kink oscillations are equally possible in the observed event.

### Discussion
The paper presents high-resolution detection, down to 123 km per pixel, of a decayless kink oscillation of a coronal loop, observed simultaneously from two vantage points by the spaceborne

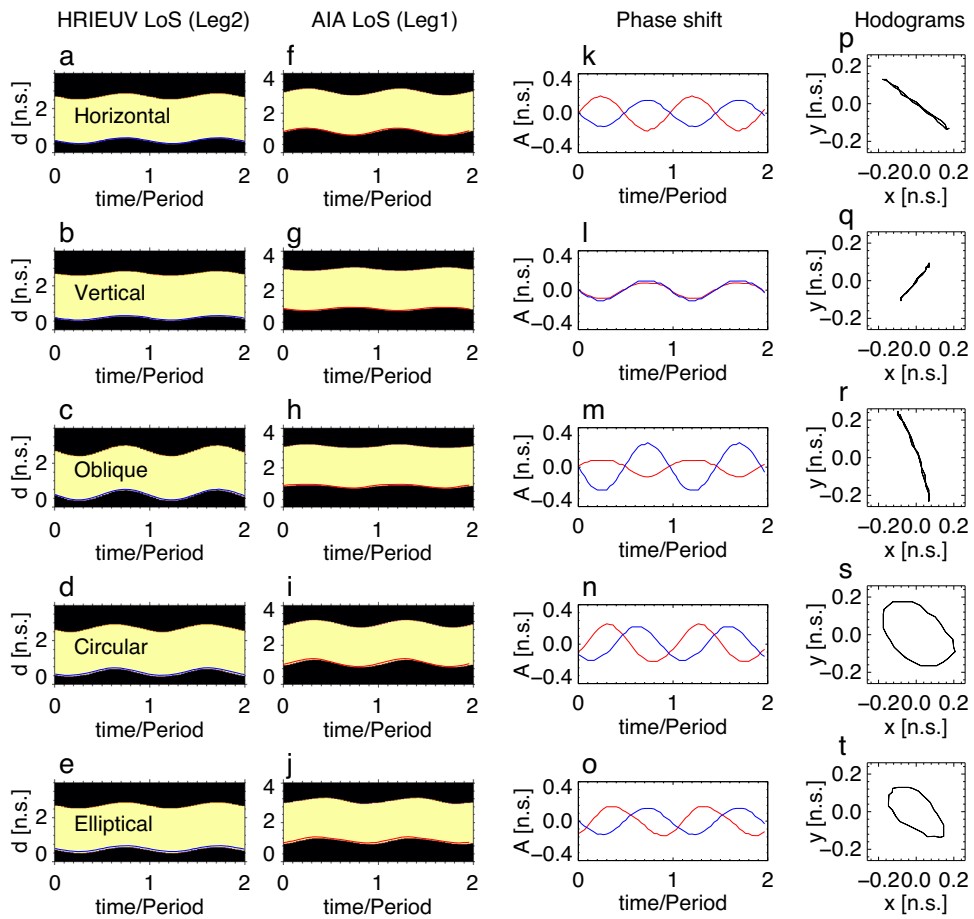

**Fig. 6 | Simulated oscillations with different types of polarisation seen in two LoS similar to that of observations. a–j** Representative time-distance maps revealing oscillations viewed in LoS mimicking HRIEUV's (**a–e**, in Leg 2) and AIA's (**f–j**, in Leg 1) view. The time-distance maps are made using slits marked by the stars in Fig. 5. The body of loop is indicated by the yellow and the background is black. The unit of length n.s. stands for normalised scale. The displacement of loop boundary are marked by the blue and red curves for the HRIEUV and AIA LoS, respectively. The oscillatory signals from two LoS are combined to reveal their correlation in Panels **k–o**. Panels **p–t** show the mutual hodograms of the oscillatory signals, where signals from AIA LoS is displayed in x-axis while HRIEUV in y-axis. Source data are provided as a Source Data file.

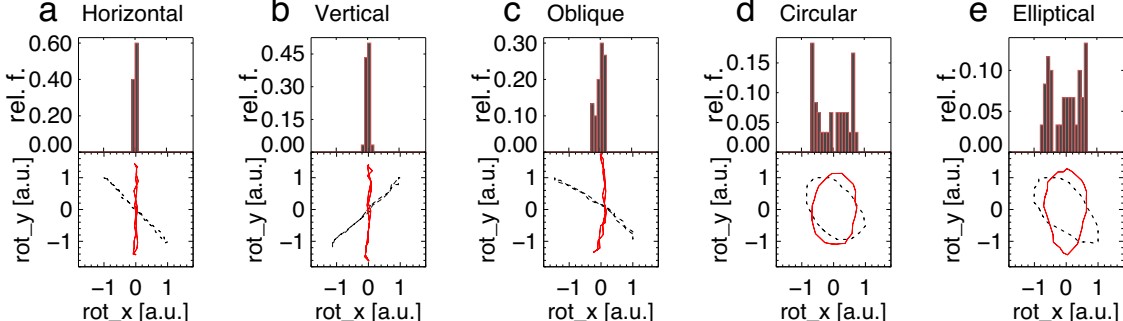

**Fig. 7 | Forward modelling the distribution of projected phase trajectories of five types of kink oscillation polarisations, seen in two LoS.** The linear horizontal (**a**), vertical (**b**), and oblique (**c**), and circular (**d**) and elliptical (**e**) polarisations are considered. The phase trajectories (black) are rotated to a certain angle to make its major axis vertically located (red). The oscillation amplitudes are normalised to their maxima. Here, rel. f. stands for Relative Frequency, a.u. stands for arbitrary units. Source data are provided as a Source Data file.

EUI/HRIEUV and AIA EUV imagers onboard the Solar Orbiter and SDO missions, respectively. The oscillatory signals detected by HRIEUV and AIA, i.e., projected on differently oriented PoS, are found to be nearly anti-phase. The mutual hodograms of the signals are highly elliptical, with a ratio of the major and minor axes of the ellipses of 0.11.

To interpret the observations and to determine the type and plane of the oscillation polarisation, we designed an analytical model mimicking the manifestation of the oscillations of certain polarisaton: the linear polarisation with different orientations of the oscillation plane; and elliptical and circular polarisations, in the PoS of HRIEUV and AIA.

We conclude that the detected kink mode is linearly polarised with the polarisation plane being highly horizontal, i.e., it is either horizontal or oblique with a strong horizontal component.

This finding has important implications for the problem of the energy balance in the solar corona. Kink oscillations of coronal loops, in the large-amplitude decaying regime, are observed to damp in a few oscillation cycles, i.e., the damping mechanism is very effective. The mechanism is linked with the effect of resonant absorption. Essentially, this effect is linear, and hence should work in low-amplitude oscillations too. But the ubiquitous nature of decayless oscillatory transverse displacements of coronal plasma loops suggests that a persistent mechanism is required to counteract processes responsible for the oscillation decay. The mechanisms proposed to compensate the energy losses in a loop which is not excited by any flare or eruption, are the random movements of footpoints or the interaction of the loop with quasi-steady flows. The mechanism based on KHI requires the oscillation to be excited somehow, which often is not observed. The mechanism based on existence of a periodic driver, e.g., the leakage of 5-min or 3-min oscillations typical for the lower atmospheric layers is excluded because of the established dependence of the decayless oscillation period on the length of the oscillating loop[17], and the lack of resonant peaks in the distribution of the detected oscillation periods.

### Randomly-driven model vs. self-oscillatory model

Since the oscillation is a linear superposition of displacements in the horizontal and vertical directions, the polarisation depends on the phase difference and amplitude difference of these two components. If the decayless oscillation is sustained by random footpoint movements, the movements are likely to be randomly directed in the horizontal plane. In other words, the horizontal motions are not likely to have a prescribed direction which could be anyhow connected with the orientation of the plane of the loop. According to the analytical model of decayless kink oscillations sustained by random footpoint motions, the resulting amplitudes in both radial and azimuthal directions are random functions, characterised by a power spectrum with peaks at loop eigen frequencies[19,32]. As the addressed problem is linear, the radial and azimuthal motions are decoupled. Likewise, the motions associated with two perpendicular linear polarisations are decoupled too, and their superposition is random. That is to say, in the randomly-driven model, the oscillation displacement is random in both time and space domains, leading to a random, time-varying phase shift between the horizontal and vertical components. Thus, the polarisation is random too. This picture is inconsistent with the observed behaviour − the constant phase shift in 11 consecutive oscillation cycles. Vertical random motions, i.e., random upflows or downflows, may sustain only vertically polarised kink oscillations as the centrifugal force is directed along the major radius of the loop, e.g., ref. 46, which does not match the observations either.

On the other hand, the self-oscillatory scenario based on the interaction of the loop with steady flows[21,23] that are perpendicular to the loop plane can readily explain the observed stable horizontal (or an oblique, but with a large horizontal component) polarisation. In 3D MHD simulations of a loop with a steady flow across either the footpoint[47] or the entire loop[23], the displacement of the loop is found to be in the same direction as the flow, i.e., linearly polarised. Note that the effect of the loop curvature with different polarisations on the oscillation frequency is minor[48,49]. It can also be illustrated by transverse self-oscillations of a violin string, excited and sustained by a steady motion of a bow. The oscillations loose energy to sound waves, while their amplitude, i.e., the sound volume, could remain at the same level for a number of oscillation periods. The energy lost by the sound wave radiation, aerodynamic friction, and friction at the pegbox and bridge, is compensated by the negative friction between the string and the bow. Importantly, the oscillation is essentially linearly polarised in the direction along the bow motion.

As demonstrated in this paper, the combination of AIA and HRIEUV at quadrature can provide unique observations to constrain the drivers of decayless observations and hence indicate the nature of the energy source sustaining the hot corona. In this context, such observations will have the maximum potential if the following criteria are fulfilled: (1) The detected loop is highly-contrasted and hosts observable oscillations. (2) The LoS angle separation between EUI and AIA should be around 90°, which happens about twice per year. (3) Simultaneous detection of the oscillating loop in the plane-of-the sky, i.e., the image should not be contaminated with other coronal structures overlapping with the loop along the LoS for both the instruments.

### Implication on coronal energy balance

As observations of decayless kink oscillations shared the same characteristics[17,50], i.e., there are no other populations related to different statistical properties, the observed case in favour of self-oscillatory model could be representative for the driving mechanism of this phenomenon.

The self-oscillatory scenario implies that the external plasma motions responsible for the decayless regime change at a characteristic time scale much longer than the oscillation period. In other words, the main energy is in the low-frequency part of the spectrum, such as, in particular, in the case of red or pink noise. Dynamic processes of this kind are typical for the solar atmosphere[51,52]. In addition, in 3D MHD simulations of the quiet Sun and coronal arcade, the time scale of coronal energy transport and release is expected to be 20–50 min[53], i.e., about an order of magnitude longer than typical oscillation periods. In the context of the coronal energy balance, this finding indicates that coronal loops can readily obtain the energy from the low-frequency, high-energy spectral band of the solar atmospheric dynamics, well outside the resonant kink frequencies of the loops. As kink oscillations are not leaky, the energy remains inside the loops until it is converted into the internal energy of the coronal plasma. The total energy budget of an oscillating loop could be inferred from the combination of the kinetic energy of kink oscillations counteracted by damping, free magnetic energy, non-thermal energy of accelerated particles, etc. From the current observation, we estimate the kinetic energy of the observed decayless kink oscillations to be as tiny as $10^{22}$–$10^{23}$ ergs. However, this figure does not correspond to the energy going into heat, as it is rather a surplus between total energy gains and resonant damping losses in the loop. The dependence of the estimated energy on the oscillation properties such as velocity amplitude could be more complicated, as in the violin scenario, for example, the friction between the string and the bow depends not only on the bowing velocity[54,55]. According to the findings of[29], this estimation is a very low limit of the actual energy converted into heat by this phenomenon. Since decayelss kink oscillations are common in the corona, with such continuous energy supply from photosphere and steady energy conversion, their contribution to the coronal energy balance could be considerable.

## Methods
### Data processing and analysis

The imaging data from HRIEUV have a time cadence of 10 s, pixel size of 0.492 arcsec, and FoV of 2048 × 2048 pixels. The HRIEUV data is internally aligned using a cross correlation technique[56] to remove the spacecraft jitters. The co-temporal AIA images in 171 Å have a time cadence of 12 s and pixel size of 0.6 arcsec. The HMI LoS magnetograms have spatial resolution of 0.5 arcsec/pixel.

The same oscillating loop in two data sets is identified based on the surrounding common features including the magnetic connectivity. Additionally, `scc_measure.pro`, a tie-pointing SolarSoft procedure is implemented to double-confirm the identity. With the help of it, the leg ahead in the HRIEUV images is found to be the right leg in the AIA images.

The motion magnification technique is a robust tool to help detecting low-amplitude quasi-periodic transverse movements in the PoS in imaging data whose spatial resolution does not allow visual inspection by naked eye. The technique we employed is based on the 2D dual-tree complex wavelet transform. The method magnifies the displacement amplitude linearly by a user-defined factor, without changing the oscillation periods[41]. The algorithm is briefly summarised as follow. First, the input images are decomposed by the dual-tree complex wavelet transform into different scales of high-pass images and low-pass images, and the relative phases of each pixel of high-pass images are calculated. Then the phase trend is smoothed over a user-defined width which is expected to be longer than the movement period. After that, the detrended phase, i.e., the phase with the trend subtracted, is multiplied by the magnification factor. Finally, the sum of the phase trend and magnified detrended phase are delivered to synthesise the magnified image sequence via the inverse dual-tree complex wavelet transform. For AIA data sets, we apply this technique to magnify the tiny transverse periodic motions. The magnification factor is within 5–10 depending on the original and desired amplitude.

To analyse the oscillations, a time–distance analysis routine is applied. In each data set, the oscillating loop is first best-fitted with a truncated ellipse. Then several slits perpendicular to the loop segments are put evenly along the ellipse to record the transverse profile at each time frame, and then time-varying profiles are tiled into a time–distance map. The slit is of 5 pixels in width, and in each frame, the profile is averaged over 5 pixels to reduce noise. In time–distance maps, the oscillatory patterns at a particular location are revealed. To track the oscillations, a transverse intensity profile of the oscillating loop (or its derivative) at each instance of time is best fitted with a Gaussian to obtain the location of the loop centre (or boundary). In this way, the time variation of the displacement of a certain loop segment is obtained. Further, the background trend is obtained by smoothing the original time series of the oscillatory displacement with a window slightly longer than the period. Then the trends are subtracted from the original signals. Oscillation periods are estimated via the Fast Fourier transform.

As mentioned in results subsection spacecraft constellation, the angle separation between SolO and SDO is 103.95 degrees, constructing a quasi-quadrature configuration. Thus, oscillations from these two data sets can be treated as two almost perpendicular components, whose trajectories in the phase plane construct a phase portrait (also known as hodogram) to reveal their phase and frequency differences. Similar to a Lissajous figure, the shape of such a hodogram carries information about the polarisation of the observed oscillations. Generally, a line, a circle, and an ellipse, is expected for linear, circular, and elliptical polarisation, respectively. In our work, we assign the normalised displacement amplitude of AIA and HRIEUV as $x$ and $y$ component respectively, and make a plot of $y$ as a function of $x$, see Fig. 4b–e.

### The geometrical model

To interpret the observed oscillation properties, we set up a geometrical model for a 3D coronal loop exhibiting fundamental kink oscillations with different types of polarisation.

The loop (boundary) is described as a semi-torus in 3D space with Cartesian coordinates[57],

$$x = (R + r \cos \phi) \cos \theta, \tag{1}$$

$$y = (R + r \cos \phi) \sin \theta, \tag{2}$$

$$z = r \sin \phi, \tag{3}$$

where $R = 15$ and $r = 1$ are the normalised major and minor radii of the loop, $\theta = [0, \pi]$ is the angle along the loop axis, and $\phi \in [0, 2\pi]$ is the angle in the azimuthal direction.

In this work, five types of polarisation with respect to the loop's plane $(x, y)$ are considered: horizontal polarisation, vertical polarisation, their linear in-phase combination with the same amplitudes as oblique polarisation, the same amplitudes but 90 degrees out of phase as circular polarisation, 90 degrees out of phase but with different amplitudes as elliptical polarisation. The periodic transverse displacements of the loop boundaries by kink oscillations are implemented by modulating the minor radius as

$$r = 1 + A_0 \sin \theta \sin(m\phi) \sin \omega t \tag{4}$$

for horizontally polarised oscillations, and

$$r = 1 + A_0 \sin \theta \cos(m\phi) \sin \omega t \tag{5}$$

for vertically polarised oscillations, where $A_0$ is the perturbation amplitude which is kept low (<0.4) for the linear regime, $m = 1$ indicates the kink symmetry of the perturbation, and $\omega t$ is the oscillation phase in radians. All other polarisation types are obtained as combinations of Eqs. (4) and (5), described above (see Supplementary Movie 2). For efficiency, the simulation runs for two oscillation cycles.

The modelled kink-oscillating loop is viewed in two particular LoS mimicking those of the HRIEUV and AIA observations considered in this work (see Fig. 5a–c). The LoS is defined by the angle $\alpha$ with respect to the loop's plane $(x, y)$, and by the angle $\beta$ to the footpoints plane $(y, z)$. Thus, for HRIEUV LoS, we use $\alpha = 1.92\pi$ and $\beta = -0.06\pi$. For AIA LoS, we use $\alpha = 0.56\pi$ and $\beta = 0.36\pi$. The angle separation between these two synthetic LoS is 109 degrees, which is about the observational one (104 degrees).

Being normal to the LoS, the planes of sky (PoS) of the HRIEUV and AIA instruments are defined by the normal vector and a fixed point (we select the midpoint between the footpoints here). Then the projections of a 3D loop onto the two PoS are obtained with standard linear algebra transformations. The 2D synthetic images of the kink-oscillating loop are thus created, with the local coordinate system redefined, so that the abscissa/ordinate is parallel/perpendicular to the line connecting the loop footpoints, respectively. The simulated 2D images are sampled with a high resolution of 0.02 unit length per pixel.

In the model described, the effects of optical depth are omitted, as we focus on the transverse displacements of the loop boundaries only (as kink oscillations are usually observed).

## Data availability

The data analysed during the current study are obtained from EUI Data Release 5.0 (https://doi.org/10.24414/2qfw-tr95) and from the Joint Science Operations Center (JSOC) database (http://jsoc.stanford.edu/). The simulation data generated in this study have been deposited in the Figshare repository (https://doi.org/10.6084/m9.figshare.23737677.v2). Source data are provided with this paper.

## Code availability

We analysed data using the Interactive Data Language (IDL), Solar-SoftWare (SSW) package, and the motion magnification technique. The routines used to process and analyse data are available at https://www.lmsal.com/sdodocs/doc/dcur/SDOD0060.zip/zip/entry/. The motion magnification code is available at https://github.com/Sergey-Anfinogentov/motion_magnification. The codes for generating modelling data have been deposited in Figshare at https://doi.org/10.6084/m9.figshare.23737677.v2.

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

## Acknowledgements

The following fundings are gratefully acknowledged: China Scholarship Council-University of Warwick joint scholarship (S.Z.), the EUI Guest Investigatorship (S.Z. and V.M.N.), STFC consolidated grant ST/X000915/1 (D.Y.K.), the Latvian Council of Science Project No. lzp2022/1-0017 (D.Y.K. and V.M.N.), the STFC Ernest Rutherford Fellowship No. ST/R004285/2 (P.A.), and the European Union funding (ERC, ORIGIN, 101039844) (L.P.C.). Views and opinions expressed are however those of the author(s) only and do not necessarily reflect those of the European Union or the European Research Council. Neither the European Union nor the granting authority can be held responsible for them. Solar Orbiter is a space mission of international collaboration between ESA and NASA, operated by ESA. The EUI instrument was built by CSL, IAS, MPS, MSSL/UCL, PMOD/WRC, ROB, LCF/IO with funding from the Belgian Federal Science Policy Office (BELSPO/PRODEX PEA 4000134088, 4000112292, 4000117262, and 4000134474), the Centre National d'Etudes Spatiales (CNES); the UK Space Agency (UKSA); the Bundesministerium für Wirtschaft und Energie (BMWi) through the Deutsches Zentrum für Luft- und Raumfahrt (DLR); and the Swiss Space Office (SSO).

## Author contributions

S.Z. performed the data analysis (both observational and modelling), produced figures, and wrote the Results and Methods sections. V.M.N. proposed the concept of the research, supervised the data analysis and modelling, and wrote the Introduction section. D.Y.K. proposed and developed the model, contributed to forward modelling and interpretation of observations. L.P.C. implemented the internal alignment for HRIEUV data. A.P. identified the analysed event in preliminary observational data. C.V. and D.B. advised on the use of EUI HRIEUV data. All co-authors contributed to the scientific discussion of the obtained results, and editing/reviewing the manuscript.

## Competing interests

The authors declare no competing interests.
