## [Peer Review File · Nature Communications]

REVIEWER COMMENTS

Reviewer #1 (Remarks to the Author):

Report on Manuscript NCOMMS-23-07737

The manuscript reports a detailed observational analysis of decayless transverse loop oscillations observed from two viewpoints by spacecraft. The use of two vantage points enables to characterise the polarisation of the motions, which is found to be horizontal or weakly oblique linear. The result is supported by forward-modelling of simulated 2D images that reproduce well the observed geometry. The claim is made that the revealed polarisation favours the sustainability of oscillations by quasi-steady flows which may supply the energy for coronal heating.

I find the results of great interest to the solar community. The motivation, methodology, data analysis and interpretation are sound. The narrative is clear, concise and to the point. The overall quality of the manuscript is excellent. However, the wider implications for coronal heating, as alleged in the introduction and discussion, are not demonstrated in the manuscript. In particular:

- The arguments in Section 3 (p.11) are mostly based on conjectures about the polarisation properties generated by different drivers, not actually based on the analysis of the response of a loop system to alternative driving properties. The main argument is illustrated with a musical example, but is not actually demonstrated with calculations or reference to published literature.

- With only one example it is difficult to generalise the finding. The polarisation of the motions is likely to depend on other factors, such as the geometry of the loop. In the absence of a general pattern derived from observations involving more cases, it is difficult to establish a firm conclusion about the implication of the finding for the excitation mechanism/coronal heating.

These two shortcomings detract from the implications of the findings as key indicators of the oscillation excitation mechanism and hence responsible for the energy supply to the corona.

Reviewer #2 (Remarks to the Author):

This is my review of the manuscript with title 'Polarisation of decayless kink oscillations of solar coronal loops'. This study deals with the important question regarding the polarisation and driving mechanism of decayless oscillation in coronal loops. The latter is a topic long debated within the community, and identifying the polarisation of such an oscillation could give us insight on the nature of its driving mechanism.

In this paper, the authors reveal, for the first time in a decayless oscillation, a horizontal or weakly oblique linear polarisation, derived from simultaneous observations by two different instruments, the SDO/AIA and the SoHO/EUI. To the reviewer's knowledge, this is a novel result with very important implications. In particular, the authors make a strong case, based on the former result, that the driving mechanism of this decayless oscillation is the interaction of the loop with steady flows, through a self-oscillatory process. This is properly explained in text, both as a concept and in relation to past analytical and numerical work in the literature. Overall, this is a well-written manuscript that is efficiently conveying the results of the current study, presenting a very good analysis with reasonable assumptions and no major flaws.

My main concern with this work is with one of the assumptions that the authors made during the analysis. In particular, I am referring to the assumption of considering the frequency fluctuations in the observed oscillatory signals as the result of noise. This is strongly connected to both major conclusions of this study. Below you can find the relevant sections where this is discussed in the text, and my comments.

1) Section 2.2, lines 148-150: 'The average length of the minor axes... where 0.7 is the half width of the major axis'.

Section 2.3, lines 183-187: 'For the circular polarisation... with the axes ratio 0.45 in the phase portrait'.

Section 2.4, lines 204-209: 'However, only models with... or another origin'.

Section 3, lines 226-228: 'We conclude that the detected kink mode... with a strong horizontal component'.

-- From the analytical model we see that the elliptical trajectory in the hodogram gives a ratio of minor to major axes of 0.6 for the circular polarisation and 0.45 for the elliptical polarisation, while it leads to a very narrow line-like structure for horizontal and oblique polarisations. From the observations, the authors find a minor to major axes ratio of 0.27 for the observed time series and

a ratio of 0.11 for the filtered time series. Based on the latter, and after comparing with the hodograms produced for the different polarisations in the analytical model, the authors conclude that the observed oscillation must be linearly polarised with a very strong horizontal component.

The main assumption that led the authors to their conclusion is that the fluctuations in the period of the two unfiltered signals come down to noise, and that this noise must be removed with a filter. However, this fluctuation manifests similarly in both signals, despite the two coming from different instruments with noticeably different spatial resolutions. This could possibly indicate a more intrinsic origin of those fluctuations, such as a more elliptically polarised oscillation. Such an assumption could be supported by comparing the ratio of the unfiltered signal (0.27) and the analytical model (0.45), again taking into account observational uncertainties.

I would like the authors to comment on this, and further explain why they believe this fluctuation to be caused by noise. This comment is not meant to remove from the results and the value of authors conclusions. In fact, their assumption about the period fluctuations is very reasonable. I would just like the authors to expand upon their reasoning, and make it more clear both in the text and in their response to the referee.

2) Section 3, lines 242-251: 'If the decayless oscillation is sustained... (or an oblique, but with a large horizontal component) polarisation'.

-- This comment is directly connected to the previous major comment, and is about the second main conclusion of this study. Assuming a linear, strongly horizontal polarisation for the decayless oscillation, the authors then make a strong case supporting a self-oscillatory process through the interaction of the loop with steady flows, as the driving force behind the observed oscillation. If this conclusion is correct, then this work potentially addresses one of the main unknowns behind decayless loop oscillations. However, if the oscillation is elliptically polarised, then we cannot exclude the possibility of random footpoint movements driving the oscillation. Random footpoint motions would result in a randomly varying polarisation, but resonance of the loop to the driving should lead to an amplified response to certain frequencies (see Afanasyev et al. 2020). The resulting oscillation could still resemble an elliptically polarised one, to a certain extent. I would like for the authors to further expand upon this, either in the text or in their response to the referee.

Minor comments

1) In the right panel of the Supplementary Movie 1, the date as mentioned in the title of that panel should have been "2022-04-01" rather than "2022-01-01". It would be better to have this fixed.

2) Section 1, line 45-46: 'the ubiquity of decayless kink oscillations makes them an important process for the coronal energy balance.'

-- Although it is expected from decayless oscillations to affect the energy balance in the solar atmosphere, the nature and importance of these effects are still under debate. In particular, the energy dissipation mechanism is still not well understood. For example, the concept of steady state KH instability turbulence requires higher-than the observed amplitudes of decayless oscillations to sustain the quiet solar corona (Hillier et al. 2020), or even predicts radiative cooling (see Hillier & Arregui 2019, for oscillating prominence threads). The authors are advised to at briefly mention some of the opposing views on the subject.

3) Section 1, line 59-60: 'So far, there has been only one attempt to estimate the polarisation plane stereoscopically, for a large-amplitude decaying regime'

-- In Aschwanden & Schrijver (2011), the authors used a combination of STEREO EUVI/A and SDO/AIA data to estimate stereoscopically the 3D loop shape, as well as the plane of oscillation with respect to the loop plane. They concluded that the observed loop kink oscillation was vertically polarized. The authors should consider rephrasing this and/or include this additional reference.

4) Section 2.3, line 176-177: 'situated near the loop apex (i.e., near the location of the 177 maximum oscillation amplitude).'

-- It should be specified that the two slits shown in Figure 5 and considered in Figure 6 technically belong to Leg 2 (for HRIEUV image) and Leg 1 (AIA image) respectively, despite being close to the apex. This way, the authors will avoid causing confusion among the readers, when they compare this sentenced to what is shown in Figures 5 and 6.

Like I have already mentioned, this a well written manuscript that is efficiently conveying the results of the current study. It includes a very good analysis with reasonable assumptions and no major flows. I find this work and its conclusions very interesting and I would be happy to read the re-submitted version of it, once it is uploaded.

Point-to-Point Response to Reviewers' comments

REVIEWER COMMENTS

Reviewer #1 (Remarks to the Author):

Report on Manuscript NCOMMS-23-07737

The manuscript reports a detailed observational analysis of decayless transverse loop oscillations observed from two viewpoints by spacecraft. The use of two vantage points enables to characterise the polarisation of the motions, which is found to be horizontal or weakly oblique linear. The result is supported by forward-modelling of simulated 2D images that reproduce well the observed geometry. The claim is made that the revealed polarisation favours the sustainability of oscillations by quasi-steady flows which may supply the energy for coronal heating.

I find the results of great interest to the solar community. The motivation, methodology, data analysis and interpretation are sound. The narrative is clear, concise and to the point. The overall quality of the manuscript is excellent. However, the wider implications for coronal heating, as alleged in the introduction and discussion, are not demonstrated in the manuscript. In particular:

- The arguments in Section 3 (p.11) are mostly based on conjectures about the polarisation properties generated by different drivers, not actually based on the analysis of the response of a loop system to alternative driving properties. The main argument is illustrated with a musical example, but is not actually demonstrated with calculations or reference to published literature.

REPLY: Your helpful comment is appreciated. In the revised version, we provide references to the previously published literature, to support our statement. We relate the polarisation to different driving mechanisms, and support this with additional references.

For the self-oscillatory model of decayless kink oscillations, there is a clear picture of linear polarisation of the excited oscillations. The self-oscillatory model has been first proposed by 2016A&A...591L...5N. The analogy of a transversely oscillating loop with the violin exploits the universality of wave theory. In particular, this analogy was first suggested by Hans Goedbloed in his seminal lecture course on MHD waves in early 80s. Recently, MHD numerical simulations on a 3D straight flux tube with self-oscillations have confirmed predictions made with the use of this analogy (2020ApJ...897L..35K). Specifically, the authors applied a horizontal flow around one of the footpoints with a constant velocity mimicking the supergranulation flow, leading to the periodic displacement of the loop's centre in the same direction. In another paper, 2021ApJ...908L...7K, a steady flow across the entire loop was found to form a street of vortices, which pushes the loop back and forth in the same direction along the flow, confirming the ideas based on the low-dimensional modelling (2009A&A...502..661N) and obtained in 2D (2010PhRvL.105e5004G). In both those 3D simulations, the resulting polarisations were linear horizontal. Therefore, the

previous studies yield that the self-sustained decayless kink oscillation is in linear polarisation mode.

Concerning the random driving model of decayless kink oscillations, it is described by an analytical model of a straight magnetic tube perturbed with a stationary random noise. In particular, 2021MNRAS.501.3017R (Article I) and 2021SoPh..296..124R (Article II) obtained the relationship of the random footpoint driver and the excited oscillations. It was found that if the driver displacement is a random function, (1) the excited loop displacement is also a random function, whose spectrum has peaks at the loop eigenfrequencies, see Eq. (70-71) and Figure 3 in Article I; (2) the displacement amplitude in the radial direction is of the same order as the driver amplitude, see Eq. (54) in Article II; (3) the amplitude in the azimuthal direction is prescribed by the driver amplitude too, see Eq. (58) in Article II. Based on these findings, the linear superposition of randomly driven loop displacements in the radial and azimuthal directions naturally results in the random polarisation of oscillations.

Responding to this comment, we put the description of the polarisation types of two competing models in a new sub-section “Randomly-driven model vs. self-oscillatory model” in Discussion, and we add the following statement in introduction and discussion:

In line 56: “In the decayless regime, the self-oscillatory model pictures a linear polarisation \cite{2020ApJ...897L..35K,2021ApJ...908L...7K}, while the randomly-driven model depicts a random polarisation \cite{2021MNRAS.501.3017R,2021SoPh..296..124R}.”

In line 251: “Since the oscillation is a linear superposition of displacements in the horizontal and vertical directions, the polarisation depends on the phase difference and amplitude difference of these two components.”

In line 255: “According to the analytical model of decayless kink oscillations sustained by random footpoint motions, the resulting amplitudes in both radial and azimuthal directions are random functions, characterised by a power spectrum with peaks at loop eigenfrequencies \cite{2021MNRAS.501.3017R,2021SoPh..296..124R}. As the addressed problem is linear, the radial and azimuthal motions are decoupled. Likewise, the motions associated with two perpendicular linear polarisations are decoupled too, and their superposition is random. That is to say, in the randomly-driven model, the oscillation displacement is random in both time and space domains, leading to a random, time-varying phase shift between the horizontal and vertical components. Thus, the polarisation is random too.”

In line 269: “In 3D MHD simulations of a loop with a steady flow across either the footpoint \cite{2020ApJ...900..167K} or the entire loop \cite{2021ApJ...908L...7K}, the displacement of the loop is found to be in the same direction as the flow, i.e., linear polarised. Note that the effect of curvature of loop with different polarisation on oscillation frequency is minor \cite{2004ApJ...606.1223V,2006ApJ...650L..91T}.”

- With only one example it is difficult to generalise the finding. The polarisation of the motions is likely to depend on other factors, such as the geometry of the loop. In the

absence of a general pattern derived from observations involving more cases, it is difficult to establish a firm conclusion about the implication of the finding for the excitation mechanism/coronal heating.

REPLY: We fully agree with this comment. Indeed, only rigorous statistical analysis of the polarisation of decayless kink oscillations could confidently constrain the driving mechanism. The current work presents the very first, unique observation allowing us to assess the polarisation from two vantage points. Similar events have not been studied or anyhow mentioned in the literature or at conferences. So far, we have performed a comprehensive search through all the available data of simultaneous observations with AIA and EUV (https://www.astro.oma.be/doi/ROB-SIDC-SolO_EUI-DataRelease6.0_2023-01.html) but unfortunately have not managed to find additional high-quality two non-parallel LoS observations of off-limb oscillating loops, which would allow for a similar analysis. It is not easy to satisfy the following criteria: (1) The detected loop is highly-contrasted and hosts observable oscillations.; (2) the LoS angle separation between EUV and AIA should be around 90 degrees; (3) simultaneous detection of the oscillating loop in the plane-of-the sky, i.e., the image should not be contaminated with other coronal structures overlapping with the loop along the LoS for both the instruments. Another coincidence of those conditions has not been found in the available data. Its probability could be improved by running a dedicated observational campaign, while the nearest available slots are in Spring 2024. The publication of the current paper would strongly facilitate such a campaign.

Concerning the result based only on one case study. So far, previous observations of decayless kink oscillations have been found to share the same properties, i.e., there are no other populations related to different statistical properties. It makes us to consider one example to be sufficiently representative. In addition, most of the currently thriving research topics began with a case study. For example, the very first publication of decaying kink oscillations of a coronal loop (Science 285, 862, 1999) was a case study, which opened up the whole field of coronal MHD seismology; the first observational evidence of Alfvénic wave heating the corona 2011Natur.475..477M presented only one example too, which has stimulated investigations on the role of waves in coronal heating. We believe our work to become pioneering in revealing the coronal heating mechanism via ubiquitous decayless kink oscillations and, hence, induce an avalanche of follow-up works both theoretical and observational.

Responding to this comment, we add the following statement in Discussion:

In line 278: “As demonstrated in this paper, the combination of AIA and HRIEUV at quadrature can provide unique observations to constrain the drivers of decayless observations and hence indicate the nature of the energy source sustaining the hot corona. In this context, such observations will have the maximal potential if the following criteria are fulfilled: (1) The detected loop is highly-contrasted and hosts observable oscillations. (2) The LoS angle separation between EUV and AIA should be around 90\textdegree, which happens about twice per year. (3) Simultaneous detection of the oscillating loop in the plane-of-the sky, i.e., the image should not be contaminated with other coronal structures overlapping with the loop along the LoS for both the instruments.”

In line 287: “As observations of decayless kink oscillations shared the same characteristics (citep{2015A&A...583A.136A,2023ApJ...944....8L}, i.e., there are no other populations related to different statistical properties, the observed case in favour of self-oscillatory model should be representative for the driving mechanism of this phenomenon.”

In line 308: “Since decayless kink oscillations are common in the corona, with such continuous energy supply from photosphere and steady energy conversion, their contribution to the coronal energy balance could be considerable.”

These two shortcomings detract from the implications of the findings as key indicators of the oscillation excitation mechanism and hence responsible for the energy supply to the corona.

REPLY: We hope that the reasoning presented address this concern and clarify our point.

Reviewer #2 (Remarks to the Author):

This is my review of the manuscript with title 'Polarisation of decayless kink oscillations of solar coronal loops'. This study deals with the important question regarding the polarisation and driving mechanism of decayless oscillation in coronal loops. The latter is a topic long debated within the community, and identifying the polarisation of such an oscillation could give us insight on the nature of its driving mechanism.

In this paper, the authors reveal, for the first time in a decayless oscillation, a horizontal or weakly oblique linear polarisation, derived from simultaneous observations by two different instruments, the SDO/AIA and the SoHO/EUI. To the reviewer's knowledge, this is a novel result with very important implications. In particular, the authors make a strong case, based on the former result, that the driving mechanism of this decayless oscillation is the interaction of the loop with steady flows, through a self-oscillatory process. This is properly explained in text, both as a concept and in relation to past analytical and numerical work in the literature. Overall, this a well written manuscript that is efficiently conveying the results of the current study, presenting a very good analysis with reasonable assumptions and no major flows.

REPLY: Thank you for the positive comments.

My main concern with this work is with one of the assumptions that the authors made during the analysis. In particular, I am referring to the assumption of considering the frequency fluctuations in the observed oscillatory signals as the result of noise. This is strongly connected to both major conclusions of this study. Bellow you can find the relevant sections where this is discussed in the text, and my comments.

1) Section 2.2, lines 148-150: 'The average length of the minor axes... where 0.7 is the half width of the major axis'.

Section 2.3, lines 183-187: For the circular polarisation... with the axes ratio 0.45 in the

phase portrait'.

Section 2.4, lines 204-209: 'However, only models with... or another origin'.

Section 3, lines 226-228: 'We conclude that the detected kink mode... with a strong horizontal component'.

-- From the analytical model we see that the elliptical trajectory in the hodogram gives a ratio of minor to major axes of 0.6 for the circular polarisation and 0.45 for the elliptical polarisation, while it leads to a very narrow line-like structure for horizontal and oblique polarisations. From the observations, the authors find a minor to major axes ratio of 0.27 for the observed time series and a ratio of 0.11 for the filtered time series. Based on the latter, and after comparing with the hodograms produced for the different polarisations in the analytical model, the authors conclude that the observed oscillation must be linearly polarised with a very strong horizontal component.

The main assumption that led the authors to their conclusion is that the fluctuations in the period of the two unfiltered signals come down to noise, and that this noise must be removed with a filter. However, this fluctuation manifests similarly in both signals, despite the two coming from different instruments with noticeably different spatial resolutions. This could possibly indicate a more intrinsic origin of those fluctuations, such as a more elliptically polarised oscillation. Such an assumption could be supported by comparing the ratio of the unfiltered signal (0.27) and the analytical model (0.45), again taking into account observational uncertainties.

I would like the authors to comment on this, and further explain why they believe this fluctuation to be caused by noise. This comment is not meant to remove from the results and the value of authors conclusions. In fact, their assumption about the period fluctuations is very reasonable. I would just like the authors to expand upon their reasoning, and make it more clear both in the text and in their response to the referee.

REPLY: Thank you for this comment. As, apparently, we did not express our reasoning as clearly as expected, we revised the paper accordingly. The discrimination between the linear and elliptical polarisation models is based on the distribution of phase trajectories in the hodograms, not on the Fourier filtering. In observations, the phase trajectories are found to be localised in the vicinity of $\text{rot}_x=0$ (see histograms in Fig. 4). In modelling, this localisation is reproduced for horizontal, vertical (excluded for another reason), and oblique linear polarisations only (see panels a-c in new Fig. 7), while circular and elliptical polarisations result in two-peak distributions (panels d and e in Fig. 7) which is distinctly different from those seen in observations. Therefore, the modelled circular and elliptical cases do not match the observed phase portrait and are therefore excluded. We also note that it was previously demonstrated that the random motions at footpoints do not change the periods of decayless kink oscillations significantly but result in the amplitude modulation (2022MNRAS.516.5227N).

Responding to this comment, we add one additional figure illustrating this reasoning, and discuss it after the description of the hodogram in line 190 as follows:

“Moreover, the distribution of phase trajectories along the minor axis is different for different polarisation. As shown in Fig.7 in the linear polarisation (panels a-c) including the horizontal, vertical, and oblique ones, the histograms of the phase trajectories along the minor axes are single-peaked with very narrow widths. It means most of the trajectories are localised near the axis origin. Circular and elliptical polarisations are manifested by bimodal distributions with double peaks at the sides of the minor axes (see panels d-e).”

In line 195, “Thus we consider ... and the corresponding hodogram shapes along with the localisation of phase trajectories along the minor axis as distinct characteristic signatures of different kink polarisation types, to be compared with observations.”

In addition, we’ve already stressed in line 213-215: “However, only models with horizontal and oblique linear polarisations (see Fig.7 a and c) successfully reproduce the localisation of the majority of phase trajectories near the origin in the hodogram in Fig.4. In other words, the observed Gaussian distribution of the minor axes in the right column of Fig.4 can be caused by the horizontal or oblique linear polarisation of oscillations only, whereas the finite width of the obtained Gaussian peak can be attributed to noise of the instrumental or another origin.”

Figure 7. Forward modelling the distribution of projected phase trajectories of five types of kink oscillation polarisations, seen in two LoS. The phase trajectories (black) are rotated to a certain angle to make its major axis vertically located (red). The oscillation amplitudes are normalised to their maxima.

2) Section 3, lines 242-251: 'If the decayless oscillation is sustained... (or an oblique, but with a large horizontal component) polarisation'.

-- This comment is directly connected to the previous major comment, and is about the second main conclusion of this study. Assuming a linear, strongly horizontal polarisation for the decayless oscillation, the authors then make a strong case supporting a self-oscillatory process through the interaction of the loop with steady flows, as the driving force behind the observed oscillation. If this conclusion is correct, then this work potentially addresses one of the main unknowns behind decayless loop oscillations. However, if the oscillation is elliptically polarised, then we cannot exclude the possibility of random footpoint movements driving the oscillation. Random footpoint motions would result in a randomly varying polarisation, but resonance of the loop to the driving should

lead to an amplified response to certain frequencies (see Afanasyev et al. 2020). The resulting oscillation could still resemble an elliptically polarised one, to a certain extent. I would like for the authors to further expand upon this, either in the text or in their response to the referee.

REPLY: Thanks for the comment. As explained in the previous reply, the elliptical polarisation is excluded.

Concerning the type of polarisation in the randomly-driven model, what matters is not only the frequency, but also properties of the oscillation in the spatial domain. In the randomly-driven model, the oscillation frequency does depend on the loop length, but the polarisation does not. Since the oscillation is a linear superposition of displacements in the horizontal and vertical directions, the polarisation depends on the phase difference and amplitude difference of these two components. For elliptical polarisation, the phase shift is a constant 90 degrees, which is independent of time. In the randomly-driven model, the oscillation displacement is random in both time and space domains, leading to a random, time-varying phase shift between the horizontal and vertical components. To be more specific, the displacements $\xi_{x,y}$ in those two directions could be described as

$$\ddot{\xi}_x + \omega_0 \xi_x = R_x(t),$$

$$\ddot{\xi}_y + \omega_0 \xi_y = R_y(t),$$

where $R_{x,y}(t)$ are random functions representing the footpoint motions in the horizontal plane, and $R_x(t) \neq R_y(t)$. According to 2021MNRAS.501.3017R and 2021SoPh..296..124R, the driven displacements $\xi_{x,y}$ have a random phase shift.

Minor comments

1) In the right panel of the Supplementary Movie 1, the date as mentioned in the title of that panel should have been "2022-04-01" rather than "2022-01-01". It would be better to have this fixed.

REPLY: Thanks. It is fixed.

2) Section 1, line 45-46: 'the ubiquity of decayless kink oscillations makes them an important process for the coronal energy balance.'

-- Although it is expected from decayless oscillations to affect the energy balance in the solar atmosphere, the nature and importance of these effects are still under debate. In particular, the energy dissipation mechanism is still not well understood. For example, the concept of steady state KH instability turbulence requires higher-than the observed amplitudes of decayless oscillations to sustain the quiet solar corona (Hillier et al. 2020), or even predicts radiative cooling (see Hillier & Arregui 2019, for oscillating prominence threads). The authors are advised to at briefly mention some of the opposing views on the subject.

REPLY: We add the following statement in line 46: "The kink oscillation damping is believed to be mainly via resonant absorption and/or KHI, though it is not clear whether the KHI

threshold is reached with the observed amplitudes in the decayless regime
(citep{2020ApJ...897L..13H}).”

3) Section 1, line 59-60: 'So far, there has been only one attempt to estimate the polarisation plane stereoscopically, for a large-amplitude decaying regime'
-- In Aschwanden & Schrijver (2011), the authors used a combination of STEREO EUVI/A and SDO/AIA data to estimate stereoscopically the 3D loop shape, as well as the plane of oscillation with respect to the loop plane. They concluded that the observed loop kink oscillation was vertically polarized. The authors should consider rephrasing this and/or include this additional reference.

REPLY: We've already cited this work (line 58-59 in the revised version): “In a few very rare cases when the top segments of the oscillating loops were parallel to the line-of-sight (LoS), the polarisation of large-amplitude decaying kink oscillations could be determined (2011ApJ...736..102A, 2014ApJ...797L..22K).”

4) Section 2.3, line 176-177: 'situated near the loop apex (i.e., near the location of the 177 maximum oscillation amplitude).'

-- It should be specified that the two slits shown in Figure 5 and considered in Figure 6 technically belong to Leg 2 (for HRIEUV image) and Leg 1 (AIA image) respectively, despite being close to the apex. This way, the authors will avoid causing confusion among the readers, when they compare this sentence to what is shown in Figures 5 and 6.

REPLY: Thanks for pointing this out. We modified it as “situated at Leg 2 in the synthetic HRIEUV image and at Leg 1 in the synthetic AIA image respectively.”

Like I have already mentioned, this is a well written manuscript that is efficiently conveying the results of the current study. It includes a very good analysis with reasonable assumptions and no major flaws. I find this work and its conclusions very interesting and I would be happy to read the re-submitted version of it, once it is uploaded.

REPLY: We are very grateful to the referee for this positive assessment.

REVIEWER COMMENTS

Reviewer #1 (Remarks to the Author):

The reasoning and arguments presented by the authors and the changes made to the manuscript have fully clarified my original concerns. I am happy to recommend acceptance of this manuscript for publication in Nature Communications.

Reviewer #2 (Remarks to the Author):

This is my review of the revised manuscript with title 'Polarisation of decayless kink oscillations of solar coronal loops'. This study deals with the important question regarding the polarisation and driving mechanism of decayless oscillation in coronal loops. The latter is a topic long debated within the community, and identifying the polarisation of such an oscillation could give us insight on the nature of its driving mechanism.

When it comes to my previous major and minor comments, the authors manage to properly address them. In particular, the addition of Figure 7 and the corresponding explanation added in the manuscript and in their direct response, have answered my question regarding the oscillation polarisation, and its link to "choice" of drivers.

I only have two short comments that I would like to be revisited by the authors.

Question + Reply

2) Section 1, line 45-46: 'the ubiquity of decayless kink oscillations makes them an important process for the coronal energy balance.'

-- Although it is expected from decayless oscillations to affect the energy balance in the solar atmosphere, the nature and importance of these effects are still under debate. In particular, the energy dissipation mechanism is still not well understood. For example, the concept of steady state KH instability turbulence requires higher-than the observed amplitudes of decayless oscillations to sustain the quiet solar corona (Hillier et al. 2020), or even predicts radiative cooling (see Hillier & Arregui 2019, for oscillating prominence threads). The authors are advised to at briefly mention some of the opposing views on the subject.

REPLY: We add the following statement in line 46: “The kink oscillation damping is believed to be mainly via resonant absorption and/or KHI, though it is not clear whether the KHI threshold is reached with the observed amplitudes in the decayless regime \citep{2020ApJ...897L..13H}.”

Referee comment

-- I would like to point out two things here. First of all, oscillation damping is not necessarily the same as energy dissipation. Indeed, kink oscillation damping is believed to be mainly via resonant absorption and/or KHI. However, my comment was about the energy dissipation aspect. Secondly, the model described in Hillier et al. (2020, or 2020ApJ...897L..13H) dealt with the efficiency of energy dissipation from a steady state KHI turbulence in decayless oscillations of KHI turbulent loops. It was not about whether or not the threshold for the development of the KHI is reached for the decayless regime of kink oscillations. Please, revisit my comment and your reply above, and make the necessary changes in the manuscript.

#####

Question + Reply

From Reviewer 1:

- With only one example it is difficult to generalise the finding. The polarisation of the motions is likely to depend on other factors, such as the geometry of the loop. In the absence of a general pattern derived from observations involving more cases, it is difficult to establish a firm conclusion about the implication of the finding for the excitation mechanism/coronal heating.

REPLY: ... In line 287: “As observations of decayless kink oscillations shared the same characteristics \citep{2015A&A...583A.136A,2023ApJ...944....8L}, i.e., there are no other populations related to different statistical properties, the observed case in favour of self-oscillatory model should be representative for the driving mechanism of this phenomenon.”

Referee comment

-- If I may, I have a short follow-up comment for one of those of Reviewer 1. In this case, I would argue that there are so many different proposed mechanisms of decayless oscillations reproducing the observational data, that only observations of the polarisation can tell them apart, so far. Since this is the first time such a dedicated (and singular) study has taken place, the authors should avoid any too strong statements. In the manuscript (line 287), the phrase "should be representative for the

driving mechanism of this phenomenon." is written as "could be representative for the driving mechanism of this phenomenon." The latter, in my opinion, is more appropriate in this case, and I would argue that it stays like that in the text.

#####

I would like to see the revised version of the manuscript, alongside the authors' response.

Reviewer #1 (Remarks to the Author):

The reasoning and arguments presented by the authors and the changes made to the manuscript have fully clarified my original concerns. I am happy to recommend acceptance of this manuscript for publication in Nature Communications.

REPLY 2: We are grateful for the recommendation

Reviewer #2 (Remarks to the Author):

This is my review of the revised manuscript with title 'Polarisation of decayless kink oscillations of solar coronal loops'. This study deals with the important question regarding the polarisation and driving mechanism of decayless oscillation in coronal loops. The latter is a topic long debated within the community, and identifying the polarisation of such an oscillation could give us insight on the nature of its driving mechanism.

When it comes to my previous major and minor comments, the authors manage to properly address them. In particular, the addition of Figure 7 and the corresponding explanation added in the manuscript and in their direct response, have answered my question regarding the oscillation polarisation, and its link to "choise" of drivers.

I only have two short comments that I would like to be revisited by the authors.

Question + Reply

2) Section 1, line 45-46: 'the ubiquity of decayless kink oscillations makes them an important process for the coronal energy balance.'

-- Although it is expected from decayless oscillations to affect the energy balance in the solar atmosphere, the nature and importance of these effects are still under debate. In particular, the energy dissipation mechanism is still not well understood. For example, the concept of steady state KH instability turbulence requires higher-than the observed amplitudes of decayless oscillations to sustain the quiet solar corona (Hillier et al. 2020), or even predicts radiative cooling (see Hillier & Arregui 2019, for oscillating prominence threads). The authors are advised to at briefly mention some of the opposing views on the subject.

REPLY: We add the following statement in line 46: "The kink oscillation damping is believed to be mainly via resonant absorption and/or KHI, though it is not clear whether the KHI threshold is reached with the observed amplitudes in the decayless regime \citep{2020ApJ...897L..13H}."

Referee comment

-- I would like to point out two things here. First of all, oscillation damping is not necessarily the same as energy dissipation. Indeed, kink oscillation damping is believed to be mainly via resonant absorption and/or KHI. However, my comment was about the energy dissipation aspect. Secondly, the model described in Hillier et al. (2020, or 2020ApJ...897L..13H) dealt with the efficiency of energy dissipation from a steady state KHI turbulence in decayless oscillations of KHI trubulent loops. It was not about about whether or not the threshold for the development of the KHI is reached for the decayless regime of kink oscillations. Please, revisit my comment and your reply above, and make the necessary changes in the manuscript.

#####

REPLY 2: We agree. That sentence was rewritten as: "On one hand, the nature and importance of damping mechanisms such as resonant absorption and KHI, are still under debate. In particular, the concept of coronal heating by steady state KHI turbulence requires higher-than the observed amplitudes of decayless oscillations to sustain the quiet solar corona [30]. On the other hand, ...", as suggested.

Question + Reply

From Reviewer 1:

- With only one example it is difficult to generalise the finding. The polarisation of the motions is likely to depend on other factors, such as the geometry of the loop. In the absence of a general pattern derived from observations involving more cases, it is difficult to establish a firm conclusion about the implication of the finding for the excitation mechanism/coronal heating.

REPLY: ... In line 287: "As observations of decayless kink oscillations shared the same characteristics \citep{2015A&A...583A.136A,2023ApJ...944...8L}, i.e., there are no other populations related to different statistical properties, the observed case in favour of self-oscillatory model should be representative for the driving mechanism of this phenomenon."

Referee comment

-- If I may, I have a short follow-up comment for one of those of Reviewer 1. In this case, I would argue that there are so many different proposed mechanisms of decayless oscillations reproducing the observational data, that only observations of the polarisation can tell them apart, so far. Since this is the first time such a dedicated (and

singular) study has taken place, the authors should avoid any too strong statements. In the manuscript (line 287), the phrase "should be representative for the driving mechanism of this phenomenon." is written as "could be representative for the driving mechanism of this phenomenon." The latter, in my opinion, is more appropriate in this case, and I would argue that it stays like that in the text.

#####

REPLY 2: In the most recent version of the manuscript we use the softer word, "could", in that sentence, i.e., it is the latter variant as suggested.

REVIEWERS' COMMENTS

Reviewer #2 (Remarks to the Author):

After reading the latest iteration of the manuscript and the authors' reply, I am glad to say that all of my comments have been addressed. I am therefore recommending the acceptance of this manuscript for publication in Nature Communications.

REVIEWERS' COMMENTS

Reviewer #2 (Remarks to the Author):

After reading the latest iteration of the manuscript and the authors' reply, I am glad to say that all of my comments have been addressed. I am therefore recommending the acceptance of this manuscript for publication in Nature Communications.

REPLY: Thank you to both reviewers, for reviewing the paper and offering your critical comments that help improve the manuscript.